# X-Forest: Approximate Random Projection Trees for Similarity Measurement

## Abstract

Similarity measurement plays a central role in various data mining and machine learning tasks. Generally, a similarity measurement solution should, in an ideal state, possess the following three properties: high accuracy, high efficiency in terms of speed and independence from prior knowledge. Yet unfortunately, vital as similarity measurements are, no previous works have addressed all of them. In this paper, we propose X-Forest, consisting of a group of approximate Random Projection Trees, such that all three targets mentioned above are tackled simultaneously. Our key techniques are as follows. First, we introduced RP Trees into similarity measurement such that accuracy is improved. In addition, we enforce certain layers in each tree to share identical projection vectors, such that exalted speed is achieved. Last but not least, we introduce randomness into partition to eliminate its reliance on prior knowledge. We conduct experiments on three real-world datasets, whose results demonstrate that our model, X-Forest, reaches an efficiency of up to 3.5 times higher than RP Trees with negligible compromise on its accuracy, while also being able to outperform traditional Euclidean distance-based similarity metrics by as much as 20% with respect to clustering tasks.

## 1 Introduction

### 1.1 Background and motivation

Similarity measurement is to measure the similarity between every pair of items in a given dataset. Generally, an item can be represented by a data point in the space (*e.g.,* Euclidean space). The target of similarity measurement is to generate a similarity matrix $M$ whose element $M_{ij}$ represents the similarity value between two data points: $i$ and $j$.

Similarity measurement plays a central role in data mining and machine learning, and also has practical applications in other fields such as biochemistry, biology, botany, *etc.* In data mining, similarity is a vital criterion in unsupervised clustering which is to classify objects into groups and eliminate inappropriate data Santos et al. (2013); Kushawah & Yadav (2016); Jarvis & Patrick (1973). The result of clustering can be applied in various specific fields, *e.g.,* the accurate segmentation of liver lesions Jha et al. (2010), the characterization of chemical structures and biological activity spectra Fliri et al. (2005), or for ligand identification Koch et al. (2004). In machine learning, similarity can be used in social filtering algorithms to make predictions for recommendation systems Billsus & Pazzani (1998).

Therefore, these extensive applications require the similarity measurement solution to maintain high accuracy in different datasets Ma & Manjunath (1996), and this is the *first design goal* of this paper. *The second design goal* is to achieve efficient similarity measurement.

### 1.2 Prior art and their limitations

For similarity measurement, existing works can be divided into two kinds: mathematical distance-based similarity and multi-partition based similarity. Currently the prevailing approaches belong to the first kind of solutions, such as Minkowski distance family, Fidelity or Squared-chord family, Shannon's entropy Cha (2007), Cosine similarity Irani et al. (2016), etc. These similarity measurement solutions only depend on the pairwise information (*i.e.,* partial information), but neglect the overall information, such as the dimensions, features, distribution of the dataset. Consequently, they have low versatility, *i.e.,* lacking the flexibility

to adapt to different datasets. Furthermore, they are not accurate enough, because they may not preserve the perceptual similarity (intuitive similarity) of the dataset, especially when encountered with high dimension datasets Ma & Manjunath (1996); Dasgupta & Freund (2008).

The second kind of solutions such as Multiple RP+EM Fern & Brodley (2003) and RF similarity Gray et al. (2013) overcome the shortcomings of the first kind by projecting and partitioning the data. Unfortunately, this kind of solutions often depends on priori knowledge about data distribution or data labels, which can hardly be acquired in common circumstances. Consequently, *The third design goal* of this paper is to eliminate the dependence on priori knowledge. No existing works can achieve all the design goals at the same time.

### 1.3 Our contributions

This paper aims to achieve the above three design goals at the same time. Towards the first and third goal, we introduce the Random Projection Tree (`RP` Tree) to similarity measurement. `RP` Tree is used to randomly partition a set of data points in a space into several disjoint subsets. It is well known that in an `RP` Tree, data points that are closely distributed, indicating their high level of similarity in space, are always partitioned into the same subset Dasgupta & Freund (2008). This means that `RP` Tree can achieve the first goal – high accuracy. As `RP` Tree uses random partitions, thus achieves the third goal – eliminating priori knowledge dependence.

Unfortunately, directly using `RP` Tree cannot achieve the second goal – high efficiency, because during each partition, we need to project all data points into a random vector which is time consuming (see details in Section 2). To address this problem, we propose the `X-Forest`. The key idea is to allow nodes at $i$, $i + X$, $i + 2X$... ($i$=0,1,2...) layers of the tree to share the same projection vector for partitioning. For example, let us assume that we are given a complete binary `RP` Tree with 4 layers. The root node denotes the entire dataset and each node denotes a subset of the entire dataset. For each inner node, the standard `RP` Tree needs to project all data points into one random vector. In this way, it projects each data point four times. Our solution is to let the nodes at the first and third layers share the same projection vector, and do so for the second and fourth layers. For the nodes at the third layer, we reuse the projection results of the first layer, and do so for the four layer. In this way, we project each data point twice instead of four times.

Obviously, sharing vectors sacrifices the randomness of projection vectors, further degrading the accuracy. Fortunately, the accuracy loss can be almost eliminated by using a great many Trees and carefully choosing the sharing parameter – $X$ (see details in Section 3.2).

We propose $\beta$-Similarity to record the results in `X-Forest` into a similarity matrix. It features better representing the similarity relationship between data points, which is proved by the higher accuracy in our clustering experiments.

Our key contributions are as follows:

- We introduce `RP` Trees into similarity measurement and proposed a new similarity matrix $\beta$-Similarity, which better reveals the similarity relationships between data points than traditional distance based similarity measurement.

- We propose `X-Forest` which significantly reduces the time of building procedure of `RP` Trees by sharing projection vectors.

- We conduct extensive experiments on three real datasets, and our experimental results show that `RP` Trees achieves two design goals, while `X-Forest` achieves all three desigan goals at the same time.

We have released codes in github anonymously so as to meet the demand of reproducibility sou. The mathematical proofs of our algorithm is detailed in supplementary materials.

## 2 Background and Related Work

### 2.1 Similarity Measurement

Existing solutions for similarity measurement can be classified into two categories: 1) mathematical distance based similarity and 2) multi-partition based similarity.

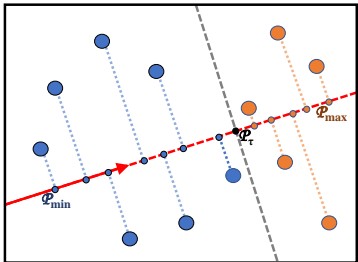
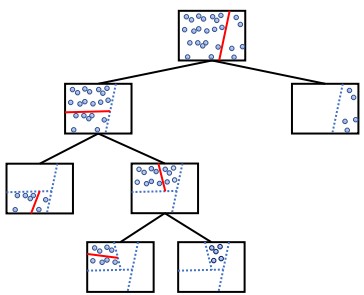

Figure 1: Relationship between projection and space partitioning. The red arrow is the projection direction. $\mathcal{P}_\tau$ the a partition point. The vertical line is the spatial partition hyperplane.

Figure 2: Example of a standard `RP` Tree. Each node uses an independent unit random projection direction.

The first kind is Mathematical distance based similarity which is widely used. It includes Minkowski distance family, Fidelity or Squared-chord and Shannon's entropy Cha (2007), Cosine similarity Irani et al. (2016), the correlation coefficient Billsus & Pazzani (1998), travel time, and edit distance Chen et al. (2009), *etc.* According to the applying method of distance, it can be divided into two cases: 1) distance is directly applied as similarity; 2) distance is first computed as a criterion for further similarity evaluation. An example of the second case is shared neighbours based clustering Jarvis & Patrick (1973). The $k$ nearest neighbours of each data point are found using Euclidean distance. The similarity between data point $i$ and $j$ is defined as the number of common neighbours.

Existing works of the second kind, multi-partition based similarity, is rare, and an example is Multiple RP+EM Fern & Brodley (2003). In each operation of RP+EM, the dimension of the original dataset is degraded through a linear transformation. Then, it applies EM clustering to generate a probabilistic model $\theta$ of a mixture of $k$ Gaussians. The similarity between data points $i$ and $j$ is defined as the average value of $P_{ij}^\theta = \sum_{l=1}^{k} P(l \mid i, \theta) \times P(l \mid j, \theta)$ of each RP+EM.

The first kind of solutions falls short in terms of leading to unsatisfied results, especially in high-dimensional spaces. And the second kind depends on prior knowledge about data labels or data distribution.

## 2.2 RANDOM PROJECTION TREE

Sanjoy Dasgupta and Yoav Freund first propose the idea of `RP` Tree Dasgupta & Freund (2008). An `RP` Tree is a variant of $k - d$ tree Bentley (1975). The most popular application of `RP` Tree is in nearest neighbours finding, where it compensates $k - d$ tree's diminishing efficacy in high-dimensional spaces Dasgupta & Sinha (2015). Other applications of `RP` Tree cover clustering Yan et al. (2009), pattern discovering Minnen et al. (2007) and nearest neighbours finding, vector quantization Dasgupta & Freund (2009), local symmetry detection in natural images Shen et al. (2016), *etc.*

The details of building an `RP` Tree are as follows. In an `RP` Tree, the root node includes all items in a given set $\mathcal{S}$. Through `RP` operation, `RP` Tree partitions $\mathcal{S}$ into two disjoint sets, each of which is a child node of the root node. The child node will be recursively partitioned until its size is smaller than a predefined threshold $\tau$. In the end, each leaf node of an `RP` Tree forms a set of size less than $\tau$. Figure 2 shows the structure of an `RP` Tree. The main operation of `RP` Tree, the `RP` operation, generates a random unit direction vector $\mathbf{e}$ for each partition. After all data points have been projected into the random direction, we uniformly choose a partition point at random within the projection range, where the projection range refers to the interval between the smallest and the largest projection value.

The formal description of an `RP` operation is as follows. `RP` Tree computes the projection value of each point in the set $\mathcal{S}$ into the *unit directional vector* $\mathbf{e}$. Let $\mathcal{P}$ be the set of projection values, *i.e.*, $\mathcal{P} = \{\mathbf{x} \cdot \mathbf{e} \mid \mathbf{x} \in \mathcal{S}\}$. Let $\mathcal{P}_{\max}$ be the maximum value in $\mathcal{P}$ and $\mathcal{P}_{\min}$ be the minimum value in $\mathcal{P}$. The partition point $\mathcal{P}_\tau$ is uniformly and randomly selected from

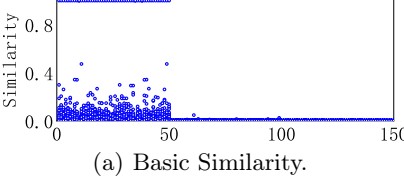
(a) Basic Similarity.

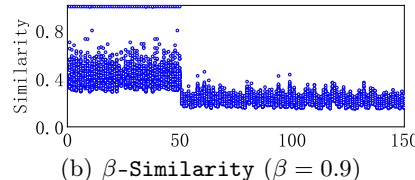
(b) $\beta$-Similarity ($\beta = 0.9$)

Figure 3: Similarity between data points of the first class (number 1-50) and the entire dataset (number 1-50, number 51-100, number 101-150).

$(\mathcal{P}_{\min}, \mathcal{P}_{\max})$. $\mathcal{P}_\tau$ partitions the set $\mathcal{S}$ into two disjoint subsets $\mathcal{S}_\text{L} = \{\mathbf{x} \cdot \mathbf{e} \leqslant \mathcal{P}_\tau \mid \mathbf{x} \in \mathcal{S}\}$ and $\mathcal{S}_\text{R} = \{\mathbf{x} \cdot \mathbf{e} > \mathcal{P}_\tau \mid \mathbf{x} \in \mathcal{S}\}$. This is equivalent to partitioning the space into two parts with a $(d-1)$-dimensional hyperplane $\mathbf{x}^T \cdot \mathbf{e} = \mathcal{P}_\tau$. When $d = 2$, as shown in Figure 1, the $(d-1)$-dimensional hyperplane degrades to a line. Building an RP Tree suffers from high computation complexity since, a large number of RP operations are involved whenever an RP Tree is built, worsened by the fact that, it requires another large number of inner products to complete a single RP operation.

## 3 The X-Forest Algorithm

As detailed in Section 2.2, RP Tree cluster similarly data points into the same node. In this section, we first show how to use a number of RP Trees to generate the similarity matrix. Then we introduce X-Forest, which significantly accelerate the building procedure of these trees.

### 3.1 Similarity Matrix Generation

In an RP Tree, two data points are regarded as similar if they are in the same leaf node. Therefore, given a number of trees that are built independently, we can consider the probability that two data points fall into the same leaf node as their similarity. Based on this idea, the well known similarity matrix is defined as follows.

**Definition 1** *Given $m$ RP Trees $\mathcal{T}_1$, ..., $\mathcal{T}_m$, for RP Tree $\mathcal{T}_i$, suppose a data point $x_j$ belongs to the leaf node $\mathcal{L}_i(\mathbf{x}_j)$ . The basic similarity matrix $\mathbf{M}^{basic}$ is defined as*

$$\mathbf{M}_{jk}^{basic} = \frac{1}{m}\left(\sum_{i=1}^{m} \mathbb{I}\left[\mathcal{L}_i(\mathbf{x}_j) = \mathcal{L}_i(\mathbf{x}_k)\right]\right) \tag{1}$$

*, where $\mathbb{I}$ is the indicator function.*

The basic similarity matrix is the average of some 01 matrix, and each 01 matrix is a similarity matrix generated by one RP Tree. In a 01 matrix, the similarity between data points falling into the same leaf node is 1 and the similarity between data points falling into different leaf nodes is 0.

This definition does not consider the information of the structure of the RP Tree. This leads to the 01 matrix is too sparse, *i.e.*, has a large number of 0s. To show that, consider the following example. Given $n$ data points to build an RP Tree with each leaf nodes containing $r$ data points, this RP Tree has $\frac{n}{r}$ leaf nodes and each leaf node fills the similarity matrix with $r^2$ 1s. Therefore, the size of this 01 matrix generated by this RP Tree is $n^2$, but only $nr$ elements are 1, accounting for $\frac{r}{n}$ of all elements (*e.g.* $n = 150$, $r = 3$, 1 accounted for only 0.02). The issue is that the similarity of data points in different leaf nodes should be a number between 0 and 1 rather than 0. Thus we can get a matrix better representing the similarity between data points, by reasonably eliminating some 0s. To achieve this, we need to consider information about the whole structure of an RP Tree. If the distance between two data points in an RP Tree is short (*i.e.*, they are divided later), we should define their similarity as a value closer to 1. If the distance between two data points in an RP Tree is long (*i.e.*, they are divided earlier), we should define their similarity as a value closer to 0. Based on this observation, we propose the $\beta$-Similarity matrix.

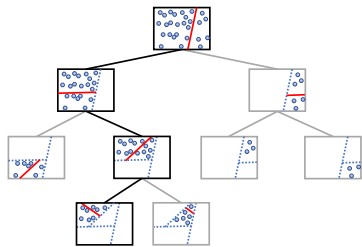 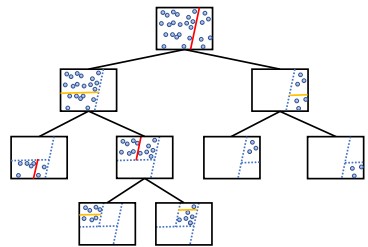

Figure 4: Example of a `layer-by-layer` RP tree. Different nodes of each layer use the same projection direction. The projection directions on each path from the root to the leaf are still independent.

Figure 5: Example of a `X-Projection` Tree. A tree uses only two random directions. One (red) for odd layers and the other (yellow) for even layers.

**Definition 2** *Given an* `RP` *Tree* $\mathcal{T}_i$, *let* $DIS_i(X, Y)$ *be the distance between nodes* $X$ *and* $Y$ *in* $\mathcal{T}_i$. *The* $\beta$-*`Similarity` matrix* $\mathbf{M}^\beta$ *is defined as*

$$\mathbf{M}^\beta_{jk} = \frac{1}{m}\left(\sum_{i=1}^{m} \beta^{\,DIS_i(\mathcal{L}_i(\mathbf{x}_j), \mathcal{L}_i(\mathbf{x}_k))}\right) \tag{2}$$

Here $\beta$ is a parameter controlling the speed at which the similarity decays with increasing distance. When $\beta \to 0$, the $\beta$-`Similarity` matrix degenerates into the basic similarity matrix. We use the most popular *Iris flower* dataset Dua & Graff (2017) to compare the basic similarity and $\beta$-`Similarity` and the result is shown in Figure 3. For basic similarity, most data points in the same class (number 1-50) have rather low similarity. For $\beta$-`Similarity`, the similarity between data points in the same class is significantly greater than that of the data points in different classes. In our experiments, we find that the accuracy of clustering can be significantly improved by properly choosing $\beta$ (Figure 8 ).

## 3.2 X-Projection Tree and X-Forest

Generating a similarity matrix requires a large number of different `RP` Trees. However, the process of building such a large number of `RP` Trees is rather slow. To address this issue, we propose `X-Forest`. The key idea is to allow nodes at $i, i + X, i + 2X, ...(i = 1, 2, ...)$ layers of the tree to share the same direction vector.

In order to introduce the `X-Forest`, we first introduce an equivalent `RP` Tree called `Layer-by-Layer` `RP` Tree. Compared to the standard `RP` Tree, `Layer-by-Layer` `RP` Tree allows nodes in each layer share the same projection direction. An example of a `Layer-by-Layer` `RP` Tree is shown in Figure 4. Theoretically, the `Layer-by-Layer` `RP` Tree is equivalent to the standard `RP` tree. Here we give a brief explanation: For a standard `RP` Tree, the partition of each node on the tree relies on a series of independent projection directions. Intuitively, a random partition of the dataset demands each data point to go through a series of mutually independent partitions. Thus, in an `RP` Tree, the nodes which are ancestor-descendant related (ADR) need mutually independent projection directions, and the nodes which are not ADR can share the same projection directions. Specifically, it is sufficient that the projection directions used by different layers are mutually independent. This shows the equivalence between the standard `RP` Tree and `Layer-by-Layer` `RP` Tree.

To further explore other avenues to economize computation time, `X-Forest` allows different layers share the same projection directions. Sharing projection directions sacrifices the randomness of partition, further affecting the similarity generated. To implement the trade-off between speed and accuracy, we give a method by adjusting the sharing parameter $X$.

Here we describe the details about building an `X-Forest`, which is a group of `X-Projection` Trees. Given a dataset $\mathcal{S}$, for each `X-Projection` Tree, we select $X$ independent random projection directions $\mathbf{e}_0, \mathbf{e}_1, \cdots, \mathbf{e}_{X-1}$, and compute the projection value of all data points into the $X$ projection directions, *i.e.* $\mathcal{P}_i = \{\mathbf{x} \cdot \mathbf{e_i} \mid \mathbf{x} \in \mathcal{S}\}$. When building an `X-Projection`

Tree, we rely on the following idea to allow data sharing of the projection directions: the root node uses the first projection direction $\mathbf{e}_0$ for partitioning. The node in the $i$-th layer uses the $(i \bmod X)$-th projection direction $\mathbf{e}_{(i \bmod X)}$, and uses the pre-calculated projection value in $\mathcal{P}_{(i \bmod X)}$ to partition the set. The rest of the recursive tree construction is identical to standard `RP` Tree. Figure 5 shows an example of using $X = 2$ version of `X-Projection` Tree to partition data points.

The `X-Projection` Tree is equivalent to a `Layer-by-Layer` `RP` Tree when its depth is no greater than $X$. When $X = 1$, it is equivalent to using a series of parallel $(d-1)$-dimension hyperplanes to partition the space. This is the most efficient case because only one `RP` operation is required. According to our experimental analysis, we find out that: 1) the $X = 2$ version of `X-Forest` achieves the best trade-off between improvement in computational efficiency and loss in partition precision. 2) the $X = 4$ version of `X-Forest` almost achieves the accuracy of `Layer-by-Layer` `RP` Trees, while requiring little additional time compared to the $X = 2$ version. Further experimental details are discussed in Section 5.

Here we provide an analysis of the computational complexity of `X-Forest`. Given a $d$-dimensional dataset $\mathcal{S}$ with $n$ data points, for $m$ `RP` Trees, the time complexity of building trees and generating the similarity matrix is $O(m \cdot n \cdot (n + \log n \cdot d))$ in the average, and $O(m \cdot n^2 \cdot d)$ in the worst case. And the complexity for `X-Forest` made of $m$ `X-Projection` Trees is always $O(m \cdot n \cdot (n + X \cdot d))$. Therefore, `X-Forest` is especially suitable for datasets satisfying $d \gg n/\log n$.

In term of implementation, `X-Forest` can calculate $\mathcal{P}_i$ in parallel for all projection directions. For the standard `RP` Tree, parallel acceleration cannot be used because information about which set of data points is to be projected into each direction remains uncertain.

## 4 APPLICATIONS OF X-FOREST

In this section, we demonstrate how to apply the similarity matrix to some classical clustering algorithms, including Kernel K-means, density clustering, and spectral clustering.

**Kernel K-means:** K-means Hartigan & Wong (1979) is the most popular unsupervised clustering algorithm. It partitions all data points into $K$ clusters by finding $K$ optimal cluster centers. The optimization goal is to minimize the sum of the distances of each data point to its nearest cluster center. The Kernel K-means is an optimization of K-means clustering. The input data points are mapped into a feature space using a nonlinear mapping $\phi$. A kernel function $\mathbf{F}_{jk} = \langle \phi(\mathbf{x}_j), \phi(\mathbf{x}_k) \rangle$ is used to calculate the distance in feature space.

For this application, `X-Forest` maps data points to unit vectors in the feature space, and the kernel function is given by the similarity matrix $\mathbf{M}$.

$$\mathbf{F}_{jk} = \langle \phi(\mathbf{x}_j), \phi(\mathbf{x}_k) \rangle = \mathbf{M}_{jk} \tag{3}$$

**Density clustering:** DBSCAN Ester et al. (1996) is the most popular density-based clustering method. In DBSCAN, a point is considered as a dense part if its $\epsilon$-neighborhood has enough points. In the process of clustering, DBSCAN arbitrarily selects an unvisited dense part and its $\epsilon$-neighborhood as a cluster, and recursively adds the $\epsilon$-neighborhoods of the dense parts already added into this cluster, until no more points can be added. This process is repeated until all dense parts are visited.

In this application, the similarity matrix $\mathbf{M}$ can be used to express the inner product of two data points in the feature space $\phi$. The distance of any two data points in the feature space can be obtained by the following formula.

$$\|\phi(\mathbf{x}_j) - \phi(\mathbf{x}_k)\|_2 = (\langle \phi(\mathbf{x}_j) - \phi(\mathbf{x}_k), \phi(\mathbf{x}_j) - \phi(\mathbf{x}_k) \rangle)^{\frac{1}{2}} = \sqrt{(2 - 2 \cdot \mathbf{M}_{jk})} \tag{4}$$

**Spectral clustering:** Spectral clustering Ng et al. (2002); Shi & Malik (2000) is an algorithm derived from graph theory and has been widely used in clustering. By defining the weight (similarity) between two data points, spectral clustering embeds data points into

an undirected weighted complete graph. The complete graph is divided into $K$ sub-graphs by cutting off the edge set with minimum weight to achieve the purpose of clustering.

Classic spectral clustering uses $\exp\left(-\|\mathbf{x}_j - \mathbf{x}_k\|^2/2\sigma^2\right)$ as the weight of the edge, where $\sigma$ is the bandwidth of the graph. In this application, we directly use $\mathbf{M}_{jk}$ of the similarity matrix $\mathbf{M}$ as the edge weight.

## 5 EXPERIMENTAL RESULTS

We first show the accuracy improvement of some classical clustering methods after using the $\beta$-`Similarity` generated by `X-Forest`. Then, we compare the performance in terms of computation time of `X-Forest` under different parameter settings.

### 5.1 EXPERIMENTAL SETUP

**Choice of Datasets:** As is shown in Table 1, we conduct experiments on three real datasets from the UC Irvine machine learning library Dua & Graff (2017), including *Wine*, *Soybean* and *WDBC*. All three datasets are labeled, allowing us to evaluate the actual performance of the clustering.

| Dataset | dimension | # categories | # data points |
|---------|-----------|--------------|---------------|
| Wine    | 13        | 3            | 178           |
| Soybean | 35        | 4            | 47            |
| WDBC    | 30        | 2            | 569           |

Table 1: A summary of datasets.

**Choice of Clustering Algorithms:** In the first part of this section, we compare the accuracy of `Kernel K-means`, `Density Clustering` and `Spectral Clustering` using $\beta$-`Similarity` generated by `X-Forest` and distance similarity. In the second part, we use `Kernel K-means` as clustering algorithm to compare the performance of `X-Forest` under different parameter settings.

**Evaluation Metrics:** We use *Accuracy* as an evaluation metric of the performance of clustering. The definition of *Accuracy* is given by the formula below. It measures the fraction of matching labels given by the clustering algorithm divided by the real label.

**Definition 3** *Let $\mathcal{S} = \{\mathbf{x}_1 : y_1, \mathbf{x}_2 : y_2, \cdots, \mathbf{x}_n : y_n\}$ be the dataset, $\hat{y}(.)$ be the label obtained by the clustering algorithm, and $\sigma(.)$ be the permutation of $n$ elements.*

$$\text{Accuracy} = \max_{\sigma}\left(\frac{1}{n}\sum_{i=1}^{n}\mathbb{I}\left[\hat{y}(\mathbf{x}_i) = \sigma(y_i)\right]\right) \tag{5}$$

**Default Configuration:** Table 2 shows the default parameter configuration of `X-Forest` on 3 data sets, and lists the average depth of the `X-Projection` trees on these datasets. Since all the average depths are between 8 and 16, we regard the $X = 16$ `X-Forest` as standard `RP` Trees and `layer-by-layer` `RP` Trees. In the experiment, we generate 1000 similarity matrices for each parameter configuration and show the average value of Accuracy and Time. All experiments are conducted on laptop with 2.6 GHZ Intel Core i7 CPU.

| Dataset | # Trees | $X$ | $\tau$ | $\beta$ | Average Depth |
|---------|---------|-----|--------|---------|---------------|
| Wine    | 1000    | 2   | 10     | 0.9     | 9.6           |
| Soybean | 1000    | 2   | 3      | 0.9     | 12.4          |
| WDBC    | 1000    | 2   | 30     | 0.9     | 15.1          |

Table 2: Default Configuraion.

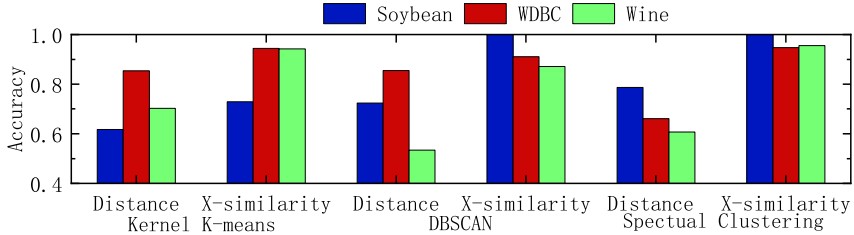

Figure 6: Accuracy of different clustering algorithms on several datasets with distance similarity and `X-Similarity`.

### 5.1.1 Experiments on Accuracy

**Accuracy vs. Different Similarity (Figure 6):** In this experiment, we use different clustering algorithms (Kernel K-means, DBSCAN, Spectral Clustring) and different similarity measurements ($\beta$-`Similarity`, distance similarity). Figure 6 shows that the Accuracy of Kernel K-means using $\beta$-`Similarity` are respectively about 11.1%, 9%, 24% higher than the of that of using distance similarity on the three datasets. The Accuracy of DBSCAN using $\beta$-`Similarity` are respectively about 27.6%, 5.5%, 33.7% higher than that of using distance similarity on the three datasets. The Accuracy of Spectral Clustering using $\beta$-`Similarity` are respectively about 21.3%, 28.6%, 34.8% higher than that of using distance similarity on the three datasets.

**Accuracy vs. $X$ (Figure 7):** In this experiment, we change the value of $X$ (1, 2, 4, 8, 16) and the number of `X-Projection` trees (20, 40, 80, 160, 320, 640, 1280) in an `X-Forest` and use Kernel K-means for clustering. When tree number is 1280, for the *soybean* dataset (Figure 7(a)), we see that the accuracy of the $X = 2$ and $X = 4$ version of `X-Forest` are respectively about 96.4% and 99.1% of that of standard `RP` Trees. For the *WDBC* dataset (Figure 7(b)), we see that the accuracy of the $X = 2$ and $X = 4$ version of `X-Forest` are almost the same as that of standard `RP` Trees. For the *wine* dataset (Figure 7(c)), we see that the accuracy of the $X = 2$ and $X = 4$ version of `X-Forest` are respectively about 99.4% and 99.8% of that of standard `RP` Trees.

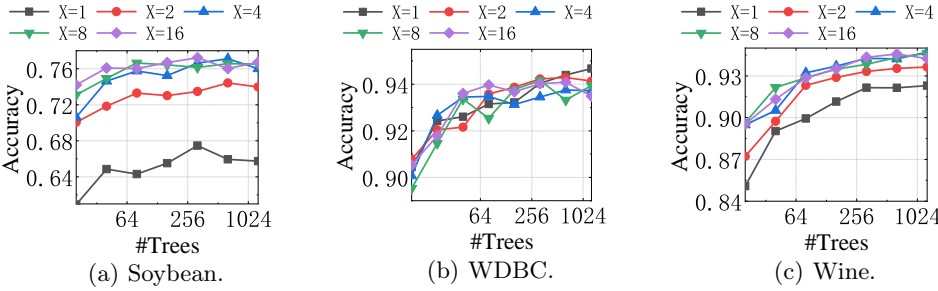

Figure 7: Accuracy of `X-Forest` with different numbers of trees and $X$s.

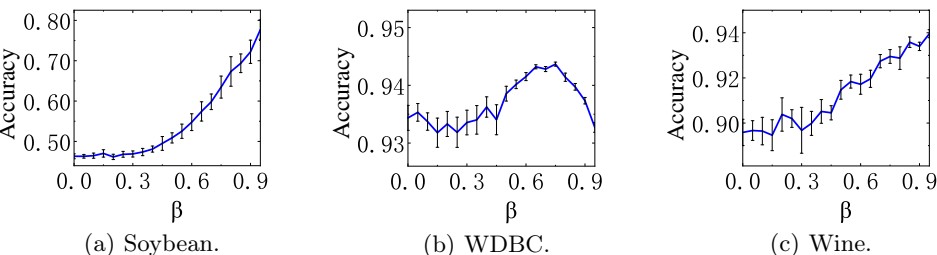

Figure 8: Accuracy and variance of $\beta$-`Similarity` with different $\beta$s.

**Accuracy vs. $\beta$ (Figure 8):** In this experiment, we change $\beta$ ($[0, 1)$) and use Kernel K-means for clustering. For the *soybean* dataset (Figure 8(a)), the accuracy of the $\beta = 0.9$ version of $\beta$-`Similarity` is about 25.9% higher than that of the $\beta = 0$ version of

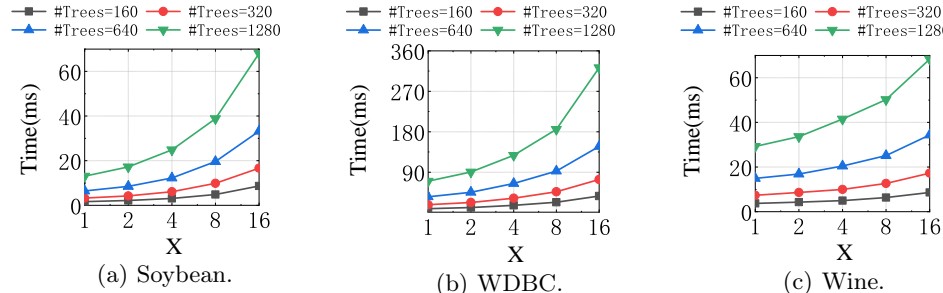

Figure 9: Time to build X-Forest for different numbers of trees and $X$s.

$\beta$-Similarity. For the *WDBC* dataset (Figure 8(b)), the accuracy of the $\beta = 0.7$ version of $\beta$-Similarity is about 0.9% higher than that of the $\beta = 0$ version of $\beta$-Similarity, while the variance is 5.28 times smaller. For the *wine* dataset (Figure 8(c)), the accuracy of the $\beta = 0.9$ version of $\beta$-Similarity is about 3.8% higher than that of the $\beta = 0$ version of $\beta$-Similarity , while the variance is 2.16 times smaller.

### 5.1.2    EXPERIMENTS ON SPEED

**Building Speed X-Forest vs. $X$ (Figure 9):** In this experiment, we vary $X$ (1, 2, 4, 8, 16) and the number of X-Projection trees (160, 320, 640, 1280) in X-Forest and use Kernel K-means for clustering. When the number of X-Projection trees is 1280, for the *soybean* dataset (Figure 9(a)), the speed of the $X = 2$ and $X = 4$ versions of X-Forest are respectively about 2.98 and 1.74 times faster than standard RP Trees. For the *WDBC* dataset (Figure 9(b)), the speed of the $X = 2$ and $X = 4$ versions of X-Forest are respectively about 3.61 and 2.57 times faster than standard RP Trees. For the *wine* dataset (Figure 9(c)), the speed of the $X = 2$ and $X = 4$ versions of X-Forest are respectively about 1.34 and 1.03 times faster than standard RP Trees.

## 6    CONCLUSION

The design goals of an ideal similarity measurement solution are respectively high accuracy, high efficiency in terms of speed and independence from priori knowledge of the dataset. We propose X-Forestto achieve all the above goals: 1) We introduce RP Tree into similarity measurement because it better represents the similarity value between item pairs; 2) We manage to reduce computational time through sharing projection values in the partition process of some layers; 3) We rely on randomness in partition to get rid of the need of priori knowledge of the dataset, such as data distribution characteristics.

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

## A  APPENDIX

The growth of the individual `X-Projection` Tree in `X-Forest` involves randomness both in projection directions and partition thresholds. A natural concern of the `X-Forest` would be whether the distances in the similarity kernel maintain the characteristics of the Euclidean space $\mathbb{R}^n$. In this appendix, we will prove that during the growth of the `X-Forest`, far-away points in the Euclidean space will be partitioned with higher probability than the near points.

Recall that during the partition of a tree node, we will choose a partition threshold randomly in the range of the projection values. As the behavior of the partition depends on this range, we first introduce the definition of the range as follows

**Definition 4** *Let $\mathcal{S}$ be a set of points. Define the range of one particular projection as the following*

$$L(\mathcal{S}, \mathbf{e}) = \sup_{\mathbf{x_1}, \mathbf{x_2} \in \mathcal{S}} \{|\mathbf{x_1} \cdot \mathbf{e} - \mathbf{x_2} \cdot \mathbf{e}|\}$$

*and define its boundaries as*

$$\rho(\mathcal{S}) = \sup_{\mathbf{e} \in \mathcal{I}} L(\mathcal{S}, \mathbf{e})$$

$$\delta(\mathcal{S}) = \inf_{\mathbf{e} \in \mathcal{I}} L(\mathcal{S}, \mathbf{e})$$

*where $\mathcal{I}$ is the set of unit direction vectors, i.e., $\mathcal{I} = \{\mathbf{e} \mid \mathbf{e} \in \mathbb{R}^n \wedge \|\mathbf{e}\| = 1\}$.*

The upper bound $\rho(\mathcal{S})$ can be seen as the largest distance of two points in $\mathbb{R}^n$, and the lower bound $\delta(\mathcal{S})$ is called *neck size* in Yan et al. (2019). Based on this definition, we will state that the probability of two points not being partitioned in a tree can be bounded by a function. Without loss of generality, we give the theorem and the proof under the 2-dimensional case with $X = 2$. When it comes to the high-dimensional, the proof can become much more difficult because projection directions are chosen on the hyper-sphere. This may be desirable for the future work.

**Theorem 1** *Let $\mathcal{S}$ be a set of points in $\mathbb{R}^2$ on which the algorithm `X-Forest` runs. Given two point $A, B \in \mathcal{S}$ having distance $l$, the probability of this two point not being partitioned in a `X-Projection` Tree after $2n$ steps satisfies*

$$\Pr(A \text{ and } B \text{ not partitioned}) \leq F_n^2\left(\frac{l}{\rho(\mathcal{S})}\right)$$

$$F_n(x) := \frac{2}{\pi(n-1)!} \int_0^{\frac{\pi}{2}} d\theta \int_{x\cos(\theta)}^1 (-\log z)^{n-1} \, dz$$

**Proof.** Let $V$ be a random variable defined as below

$$V = \begin{cases} 0, & A \text{ and } B \text{ not partitioned} \\ 1, & \text{otherwise} \end{cases}$$

Since the parameter is set as $X = 2$, two projection directions will be chosen at the beginning of the algorithm. Let $\mathbf{a}$, $\mathbf{b}$ represent them respectively. Let $Y_1, Y_2$ be random variables representing the angle between the two directions and $\mathbf{AB}$ respectively. It is obvious that the angle locates in the interval $[0, \pi/2]$. Because these vectors are uniformly and independently chosen at random from $\mathcal{I}$, we have $Y_1, Y_2 \sim i.i.d \ U_{[0,\pi/2]}$.

We first analyze the probability of $A$ and $B$ not being partitioned given particular $\theta_1$ and $\theta_2$, i.e., $\Pr(V = 0 \mid Y_1 = \theta_1, \ Y_2 = \theta_2)$. We notice that the partition performed on the direction $\mathbf{a}$ affects the partition on direction $\mathbf{b}$. This is because the partition on direction $\mathbf{a}$ will reduce the elements of $\mathcal{S}$, which shrinks the length $L(\mathcal{S}, \mathbf{b})$. If we assume that partitions on two directions are independent with each other, then this shrink will not happen. In this case, the probability of $A$ and $B$ not partitioned in a tree becomes larger. That is

$$\Pr(V = 0 \mid Y_1 = \theta_1, \ Y_2 = \theta_2) \leq \Pr(V_{\mathbf{a}} = 0 \mid Y_1 = \theta_1) \cdot \Pr(V_{\mathbf{b}} = 0 \mid Y_2 = \theta_2) \qquad (6)$$

where $V_{\mathbf{a}}$ and $V_{\mathbf{b}}$ are random variables representing whether point $A$ and $B$ are partitioned on direction $\mathbf{a}$ and $\mathbf{b}$ respectively, similar to $V$. It is obvious that $\Pr(V_{\mathbf{a}} = 0 \mid Y_1 = \theta_1)$ has the same form as $\Pr(V_{\mathbf{b}} = 0 \mid Y_2 = \theta_2)$, therefore we just need to focus on the direction $\mathbf{a}$.

Recall that when the depth of a tree node is even, direction $\mathbf{a}$ will be used. Therefore, $n$ partition thresholds will be chosen on direction $\mathbf{a}$. We can simplify the procedure of choosing $n$ partition thresholds as shrinking the total length $L(\mathcal{S}, \mathbf{a})$ by $n$ times. The probability of the final total length being bigger than $l\, cos(\theta_1)$ is the probability of the two points $A$ and $B$ not being partitioned in direction $\mathbf{a}$. In our algorithm, the partition threshold is uniformly chosen at random in the current range recursively, which means that the shrink factor satisfies $U_{[0,1]}$. Therefore, we have

$$\Pr(V_{\mathbf{a}} = 0 \mid Y_1 = \theta_1) = \Pr\left(L(\mathcal{S}, \mathbf{a}) \cdot Z_1 Z_2 \cdots Z_n \geq l\, cos(\theta_1)\right)$$
$$= \Pr\left(Z(n) \geq \frac{l\, cos(\theta_1)}{L(\mathcal{S}, \mathbf{a})}\right) \tag{7}$$

where $Z_1, Z_2, \ldots, Z_n$ are i.i.d random variables such that $Z_i \sim U_{[0,1]}$, and $Z(n)$ is the product of $Z_1, Z_2, \ldots, Z_n$.

We now derive the upper bound of the probability as the following

$$\Pr(V = 0) = \Pr\left(V \leq 0,\ Y_1 \leq \frac{\pi}{2},\ Y_2 \leq \frac{\pi}{2}\right)$$
$$= \int_0^{\frac{\pi}{2}} d\theta_1 \int_0^{\frac{\pi}{2}} d\theta_2 \cdot \Pr(V = 0 \mid Y_1 = \theta_1,\ Y_2 = \theta_2) \cdot f_{Y_1}(\theta_1) \cdot f_{Y_2}(\theta_2)$$

where $f_{Y_1}$ and $f_{Y_2}$ is the probability density function (PDF) of $Y_1$ and $Y_2$. Combining (6), (7) and the result from Lemma 2, we find

$$\Pr(V = 0) \leq \int_0^{\frac{\pi}{2}} d\theta_1 \int_0^{\frac{\pi}{2}} d\theta_2 \cdot \Pr\left(Z(n) \geq \frac{l\, cos(\theta_1)}{L(\mathcal{S}, \mathbf{a})}\right) \cdot \Pr\left(Z(n) \geq \frac{l\, cos(\theta_2)}{L(\mathcal{S}, \mathbf{b})}\right) \cdot \frac{4}{\pi^2}$$
$$\leq \frac{4}{\pi^2} \int_0^{\frac{\pi}{2}} d\theta_1 \int_0^{\frac{\pi}{2}} d\theta_2 \cdot \Pr\left(Z(n) \geq \frac{l\, cos(\theta_1)}{\rho(\mathcal{S})}\right) \cdot \Pr\left(Z(n) \geq \frac{l\, cos(\theta_2)}{\rho(\mathcal{S})}\right)$$
$$= \frac{4}{\pi^2 (n-1)!^2} \int_0^{\frac{\pi}{2}} d\theta_1 \int_0^{\frac{\pi}{2}} d\theta_2 \int_{\frac{l\, cos(\theta_1)}{\rho(\mathcal{S})}}^1 dz_1 \int_{\frac{l\, cos(\theta_2)}{\rho(\mathcal{S})}}^1 dz_2 \cdot (\log z_1 \log z_2)^{n-1}$$
$$= F_n^2\left(\frac{l}{\rho(\mathcal{S})}\right)$$

$\square$

The inequality of Theorem 1 shows that the probability of the two point not being partitioned has an upper bound $F_n^2(l/\rho(\mathcal{S}))$. However, the reader may want to know whether $F_n$ gives a reasonable bound for the probability. Therefore, we give a lemma to further illustrate the feature of $F_n$.

**Lemma 1** $F_n(x)$ *is a **strictly monotonically decreasing** function with respect to $x$ in $(0, 1)$, and satisfies*
$$F_n(0) = 1\ ,\quad F_n(1) = 0$$

**Proof.** First we prove that $F_n$ is strictly monotonically decreasing. It is obvious that $(-\log z)^{n-1} > 0$ for any $z \in (0, 1)$. Therefore, given $x_1, x_2 \in (0, 1)$ such that $x_1 < x_2$, we have

$$\int_{x_1 cos(\theta)}^1 (-\log z)^{n-1}\, dz\ >\ \int_{x_2 cos(\theta)}^1 (-\log z)^{n-1}\, dz$$

Apply the inequality above to the expression of $F_n$ yields

$$
\begin{aligned}
F_n(x_1) &= \frac{2}{\pi(n-1)!} \int_0^{\frac{\pi}{2}} d\theta \int_{x_1 cos(\theta)}^1 (-\log z)^{n-1} \, dz \\
&> \frac{2}{\pi(n-1)!} \int_0^{\frac{\pi}{2}} d\theta \int_{x_2 cos(\theta)}^1 (-\log z)^{n-1} \, dz \\
&= F_n(x_2)
\end{aligned}
$$

which indicates that $F_k$ is strictly monotonically decreasing. Now we calculate

$$
\begin{aligned}
F_n(0) &= \frac{2}{\pi(n-1)!} \int_0^{\frac{\pi}{2}} d\theta \int_0^1 (-\log z)^{n-1} \, dz \\
&= \frac{1}{(n-1)!} \int_0^1 (-\log z)^{n-1} \, dz \\
&= \frac{1}{(n-1)!} \int_0^\infty u^{n-1} \cdot e^{-u} \, du \qquad (u = -\log z) \\
&= \frac{1}{(n-1)!} \Gamma(n) \\
&= 1 \\
F_n(1) &= \frac{2}{\pi(n-1)!} \int_0^{\frac{\pi}{2}} d\theta \int_1^1 (-\log z)^{n-1} \, dz \\
&= 0
\end{aligned}
$$

$\square$

The combination of Theorem 1 and Lemma 1 shows that the probability of far-away points not being partitioned in a `X-Projection` Tree is low. This is because far-away points will have a bigger $l$, resulting in a smaller $F_n^2(l/\rho(\mathcal{S}))$ which approaches 0. On the other hand, near points will have a loose upper bound approaching 1, therefore they are more likely to stay together. In summary, we state that the characteristics of the Euclidean space are maintained in our `X-Forest` algorithm.

**Lemma 2** *Suppose $Z_1, Z_2, \ldots, Z_n$ are i.i.d random variables such that $Z_i \sim U_{[0,1]}$. Let $Z(n) = Z_1 Z_2 \cdots Z_n$. The probability density function (PDF) of $Z(n)$ is*

$$
f_{Z(n)}(x) = \begin{cases} \frac{(-\log x)^{n-1}}{(n-1)!}, & 0 < x \le 1 \\ 0, & \text{otherwise} \end{cases}
$$

**Proof.** We use mathematical induction on $n$ to prove the lemma.

**Base case.** When $n = 1$, we have $Z(n) = Z_1 \sim U_{[0,1]}$. Obviously the statement is true.

**Inductive step.** Assume the statement holds for $n = k$. For brevity, we rewrite $f_{Z(k)}(x)$ as $f_k(x)$.

First we calculate the cumulative distribution function (CDF) of $Z(k+1)$ based on the assumption. For $x \in [0, 1]$ we find

$$\Pr\left(Z(k+1) \le x\right) = \int_0^1 \Pr\left(Z(k+1) \le x \mid Z(k) = y\right) f_k(y)\, dy$$

$$= \int_0^x 1 \cdot f_k(y)\, dy + \int_x^1 \frac{x}{y} \cdot f_k(y)\, dy$$

$$= \Pr\left(Z(k) \le x\right) + x \int_x^1 \frac{f_k(y)}{y}\, dy$$

Differentiating both sides we have

$$f_{k+1}(x) = f_k(x) + \int_x^1 \frac{f_k(y)}{y}\, dy - x \cdot \frac{f_k(x)}{x}$$

$$= \int_x^1 \frac{f_k(y)}{y}\, dy$$

$$= \int_x^1 \frac{(-\log y)^{k-1}}{(k-1)!} \cdot \frac{1}{y}\, dy$$

$$= \int_{\log x}^0 \frac{(-t)^{k-1}}{(k-1)!}\, dt \qquad\qquad (t = \log y)$$

$$= \frac{(-\log x)^k}{k \cdot (k-1)!}$$

Therefore, the statement holds for $n = k+1$ when $x \in [0, 1]$. For $x < 0$ or $x > 1$, it is obvious that $f_{k+1}(x) = 0$. In summary, the lemma holds for $n = k + 1$. $\qquad\square$

