# OpenReview forum: "X-Forest: Approximate Random Projection Trees for Similarity Measurement"
_ICLR.cc/2020/Conference — Reject_

### Official Review · AnonReviewer3 · 2019-10-21
**Official Blind Review #3**

**Rating:** 1

**Review:**

The paper considers random projection forests for similarity measurements (which have been proposed earlier) and proposes to accelerate them by reusing projections. Tree levels up-to level-X use distinct random transformations, and subsequent levels cycle through existing projections (X of them). As this kind of reuse reduces the quality of trees, the paper proposes to (greatly) increase the number of trees in the forest. The paper also introduces a sensible "beta-similarity" which is based on average tree-distance between leafs into which the two data-points fall, rather than fraction of trees in which they fall into the same leaf-node.

I recommend to reject the paper -- as the contribution in my opinion is incremental, not principled (more of an engineering trick) and not very convincing. Furthermore, the paper has a number of issues with presentation, language, and experimental results. RP trees have been introduced over a decade ago (Dasgupta and Freund, 2008), and the authors cite a reference to RP forests from (Yan et al, 2019, IEEE Big data). Level-wise use of the same projection within a tree (that the authors call "layer-by-layer RP Trees") has been discussed (and shown to be effective) in the original Dasgupta and Freund paper.  The paper makes it very hard to understand what is a contribution, and what is borrowed from existing papers -- as the authors say "we introduce" both for well-known concepts (like RP trees), and for what I understand to be their contribution. I would ask to clearly state what is existing work, and what is new, and what are the key contributions.

Other comments.
1. Experiments:  you cite a paper on RP-forests (Yan et al, 2019) -- so ensembles of RP trees have already been proposed.  Why in the experiments you still compare only to a single RP-tree?  You report speed gains ~ 2x or 3x  w.r.t standard RP Trees -- even when you use very many (hundreds or thousands of trees in an X-forest).  Are these layer-by-layer RP trees, or do you use new random projection for each node? Are these your implementations of RP trees, or did you use existing code? The experiments do not provide enough details on the implementation to judge their significance -- e.g. 2x gain of speed could be achieved by better software implementation of the same algorithm.  Is it standard to measure clustering quality by  taking a classification problem and assuming that clusters should correspond to class labels? There are various other measures of cluster quality that do not rely on labels (e.g. rand-index, homogeneity, ratio of within-cluster to inter-cluster distances e.t.c.).

2. In figure 8, (a),(b),(c) -- it looks like choice of beta can have a dramatic impact on accuracy, and can either be monotone, or peak at intermediate values.  How do you propose to chose beta in practice -- don't you need supervision?  What is the cost to compute beta-similarity compared to naive 0-1 similarity?

3. The claim that similarity measures should be independent from prior knowledge -- can be controversial -- and is not a widely accepted truth. The field of metric learning tries to find better similarity matrices based on additional prior information. It's hard to imagine that a single similarity matrix will be suitable for all problem domains. I agree that it's useful to have a generic default similarity measure to try first, but clearly if prior data is available it should be used.

4. In addition to "mathematical distance-based similarity" and tree-based similarities -- there is also considerable work on similarity based on non-linear embeddings -- e.g. T-SNE or UMAP, provide an embedding, and then a simple cosine similarity can be used after the embedding. Overall -- this is a complex nonlinear similarity measure, not captured by your two classes.

5. A well known paper "Extremely randomized trees" also proposes kernels (similarity measurements) based on a forest created with (nearly) random splits, and should be cited, and used in comparisons. There is even a scikit-learn implementation -- called random trees embedding.

6. The introduction of the paper:  "Similarity measurement is to measure similarity..." conveys no information. The entire first paragraph does not have much content, and should be rewritten or skipped.  Bad word choice:  Unsupervised clustering is "to classify", "Exalted speed", dimension of original dataset is "degraded"...   If you say "we introduce RP Trees" -- it sounds like you propose them in this paper -- which is clearly not the case, as you cite Dasgupta and Freund's paper.   You can instead say -- we consider RP trees introduced by (ABC)...


**Experience Assessment:**

I have read many papers in this area.

**Review Assessment: Checking Correctness Of Derivations And Theory:**

I assessed the sensibility of the derivations and theory.

**Review Assessment: Checking Correctness Of Experiments:**

I assessed the sensibility of the experiments.

**Review Assessment: Thoroughness In Paper Reading:**

I read the paper thoroughly.

---

### Official Review · AnonReviewer2 · 2019-10-22
**Official Blind Review #2**

**Rating:** 3

**Review:**

This paper proposes the similarity measure called 'beta-similarity' generated by an ensemble of Random projection trees (RP trees) by Dasgupta & Freund (2008). To reduce the computational costs for building many RP trees, the paper develops an efficient approximate version called X-Projection trees by first generating X independent random projection directions, and then by sharing them at layers in turns. X-Forest is a set of X-Projection trees with X different random projections, and the proposed 'beta similarity' between x and y is defined by distances between a leaf region having x and one having y in PR trees. Experimental evaluations demonstrated that the use of beta similarity improves the clustering accuracy using it within three types of methods (kernel k-means, DBSCAN, Spectral clustering).

The paper's idea of defining a similarity using many RP Trees with different randomization is quite interesting and sounds promising given that the recently reported performance of tree ensembles such as XGBoost and LightGBM is very good. Partition-based trees plus randomizations are known to have very nice properties particularly in high dimensions, theoretically speaking.

However, this paper also has several problems 1) novelty and 2) confusing and imprecise statements.

1) novelty

The novelty of the paper is basically (a) a simple computation savings of X-forest and (b) a definition of beta similarity.

For (a), the novelty is rather small, and how much this computation savings have a practical impact is questionable since reported computation timings in Figure.9 are in ms. Furthermore, trivial parallelization would be possible because individual computations of RP trees in the ensemble are independent. Also, there exists a highly cited paper by Yan et al KDD'09 proposed a fast clustering method based on RP trees as "fast approximate spectral clustering" in their title.

On the other hand, (b) would be novel but any consideration about alternatives is not given, and the definition sounds quite heuristic and less convincing.

RP trees are existing spatial structures (proposed by Dasgupta & Freund) extending widely used k-d trees. It naturally defines the spatial closeness of data points, and thus the use of RP trees to define the similarity, and applying them to clustering (kernel k-means, DBSCAN, spectral) is not new. RP trees were proposed as an alternative of k-d trees, and the primary applications would be for nearest neighbor search or data compression like vector quantization.

2) confusing and imprecise statements

Many confusing and imprecise statements exist.

2a) The proposed 'beta similarity' eq (2) seems to lack the definition of DIS_i(X, Y). It would be something like the path length between node X and Y, or steps to the LCA (the lowest common ancestor). Also, the number m (the number of trees?) is also undefined.

2b) The three goals are set: accuracy, efficiency, and independence from prior knowledge. But when we use RP trees, 1st and 3rd are considered as resolved and sounds like the only remaining problem is 'efficiency' for their computations. Also, the third goal 'independence from prior knowledge' is quite vaguely explained, and hard to understand. For example, the affinity matrix in spectral clustering or kernel matrix with an RBF kernel is the case? It would be a kind of hyperparameters but not like 'dependence on prior knowledge'.

2c) Also, 'accuracy' of 'similarity measurements' is quite ambiguous. The use of kernel distance with RBF kernel is less accurate than the use of RP trees?? The "similarity measurement" sounds like the problem of definition, and it cannot be accurate or inaccurate. The distance-based similarities themselves have no problems, and even when RP trees are used, we need some distance metric (i.e. Euclidean distance) in a space.


**Experience Assessment:**

I have read many papers in this area.

**Review Assessment: Checking Correctness Of Derivations And Theory:**

I did not assess the derivations or theory.

**Review Assessment: Checking Correctness Of Experiments:**

I carefully checked the experiments.

**Review Assessment: Thoroughness In Paper Reading:**

I read the paper thoroughly.

---

### Official Review · AnonReviewer1 · 2019-11-04
**Official Blind Review #1**

**Rating:** 3

**Review:**

This paper proposes a new method for measuring pairwise similarity between data points. The idea is to define the similarity between two data points to be the probability (over the randomness in constructing the trees) that they are close in an RP tree. More concretely, the proposed method constructs a collection of RP trees (albeit with some modifications), and takes the similarity to be the average over different RP trees of a strictly decreasing function of the distance between the leaf nodes containing the data points in each RP tree. The key modification to the RP tree is to limit the number of projection vectors used in an RP tree and re-use previous projection vectors.

I believe the method is more or less equivalent to Euclidean distance, for the following reason. Two data points would have the highest similarity under the proposed similarity measure if they are in the same leaf node. For this to happen, both points must be on the same side of the dividing hyperplane corresponding to each of the ancestor nodes. Because the threshold along the projection vector is chosen randomly at uniform, this means that for the two data points to have high similarity consistently, the distance between them along the projection vector must be small (so that the probability of splitting them is small). Because the projection vectors themselves are chosen randomly on the unit sphere, this implies that this must hold along most projection vectors for similarity to be high consistently, which means that the Euclidean distance between the two points must be low.

If true, this raises several questions:

1) Why does the proposed method work better than distance similarity (which I assume means Euclidean distance) in the experiments? Are there situations when the proposed method would yield a high similarity consistently whereas Euclidean distance wouldn’t? Are the results in Fig. 6 just for a single run of the proposed method? If so, many more runs need to be performed since the decisions of the RP tree should vary significantly depending on the projection vectors and thresholds. Both the mean and standard deviation should be reported.
2) In Sect. 1.2, the authors critiqued distance-based similarity because it often does not correspond to intuitive notions of similarity/perceptual similarity. However, it does not appear that the proposed method would correspond to perceptual similarity either. For example, consider a dataset where some coordinates are more perceptually important than others (this is the case for example for the wavelet coefficients of a natural image - the lower frequencies are typically more perceptually important than higher frequencies). A more perceptually meaningful distance than Euclidean distance would be a Mahalanobis distance (which can essentially weight different coordinates differently), but the random projections use standard inner products and so are unable to capture the appropriate weighting of the different coordinates. So, why would one expect the proposed method to be more perceptually meaningful?
3) In Sect. 1.2, the authors critiqued multi-partition-based similarity because it does depend on the data distribution and cited the elimination of “prior knowledge dependence” in Sect. 1.3 as one of the benefits of the proposed method. This appears to be at odds with the goal of devising a similarity measure that is perceptually aligned, because such a similarity measure must depend on the representation of the data (e.g.: if the data is represented in the wavelet domain, one needs to know which order the different dimensions are arranged, i.e. from lowest frequency to highest frequency or the other way around).

Overall, it is unclear if the desiderata makes sense, and if the proposed method achieves the objectives.

Other questions:

4) For the X-Projection tree (which re-uses projection vectors), it seems to be equivalent to a layer-by-layer RP tree with larger branching factor. If so, the presentation of the method should be changed to this, because a layer-by-layer RP tree with larger branching factor is both conceptually clearer and simpler to analyze. If not, the proposed method should be compared to a layer-by-layer RP tree with larger branching factor, to justify the increased conceptual complexity of the X-Projection tree.
5) For the experiments, comparisons should also be made to multi-partition-based similarity, like Multiple RP+EM and RF similarity.
6) In addition, the proposed method should be compared to two simpler baselines that computes the average and the minimum distance of the two points along multiple random projection vectors, in order to justify the increased conceptual complexity of RP trees.
7) In Sect. 1.3, the paper claims that “it is well known that in an RP Tree, data points that are closely distributed, indicating their high level of similarity in space, are always partitioned into the same subset”. This is not true, since hyperplane could divide a cluster down the middle for example.
8) One of the claimed contributions in the abstract that “we introduce randomness into partition to eliminate its reliance on prior knowledge”. Note that just by introducing randomness, prior knowledge is not necessarily eliminated. For example, the way in which random projection is performed (i.e. standard inner product vs. other inner products) assumes knowledge of the distance metric, which is induced from the inner product.
9) In Fig. 6, what is the distance metric used for the baseline?

Minor issues:

pg. 2: “project all data points into one random vector” -> “project all data points along one random vector”
pg. 3: “leading to unsatisfied results“ -> “leading to unsatisfactory results”
pg. 3: “nearest neighbours finding” -> “nearest neighbour search”
pg. 3: “pattern discovering” -> “pattern discovery”
pg. 4: “similarly data points” -> “similar data points”
pg. 4: “01 matrix” -> “0-1 matrix”

**Experience Assessment:**

I have published one or two papers in this area.

**Review Assessment: Checking Correctness Of Derivations And Theory:**

N/A

**Review Assessment: Checking Correctness Of Experiments:**

I carefully checked the experiments.

**Review Assessment: Thoroughness In Paper Reading:**

I read the paper thoroughly.

---

### Decision · Program_Chairs · 2019-12-19

**Decision:**

Reject

**Comment:**

This paper proposes a new method for measuring pairwise similarity between data points. The method is based on the idea that similarity between two data points to be the probability (over the randomness in constructing the trees) that they are close in a Random Projection tree.

Reviewers found important limitations in this work, pertaining to clarity of mathematical statements and novelty. Unfortunately, the authors did not provide a rebuttal, so these concerns remain. Moreover, the program committee was made aware of the striking similarities between this submission and the preprint https://arxiv.org/abs/1908.10506 from Yan et al., which by itself would be grounds for rejection due to concerns of potential plagiarism.
As a result, the AC recommends rejection at this time.